# Research on Crossing-Pipe Support Structure Defect Detection of EMAT-Excited CSH Wave

**DOI:** 10.3390/s23125535

**Published:** 2023-06-13

**Authors:** Yang Hu, Jinjie Zhou, Wenying Yue

**Affiliations:** 1School of Mechanical Engineering, North University of China, Taiyuan 030051, China; hy15536307062@163.com (Y.H.); 13015480710@163.com (W.Y.); 2Shanxi Key Laboratory of Intelligent Equipment Technology in Harsh Environment, Taiyuan 030051, China

**Keywords:** periodic permanent magnet electromagnetic acoustic transducer, circumferential shear horizontal wave, pipe support, defect detecting

## Abstract

A circumferential shear horizontal (CSH) guide wave-detection method using a periodic permanent magnet electromagnetic acoustic transducer (PPM EMAT) was proposed to solve the defect detection located at the inside of the pipe welded by supporting structures. Firstly, a low-frequency CSH_0_ mode was selected to establish a three-dimensional equivalent model for the defect detection to cross the pipe support, and the ability of the CSH_0_ guided wave to propagate through the support and weld structure was analyzed. Then, an experiment was used for the further exploration of the influence of different sizes and types of defects on detection after using the support, as well as the ability of detection mechanism to cross different pipe structures. The results show that both the experiment and the simulation received a good detection signal at 3 mm crack defects, which proves that the method can detect the defects by crossing the welded supporting structure. At the same time, the support structure shows a greater impact on the detection of small defects than the welded structure. The research in this paper can provide ideas for guide wave detection across the support structure in the future.

## 1. Introduction

As the principal way of transporting oil and natural gas, the pipeline is a key piece of equipment within electric power, water power, heat power, nuclear energy, and the petrochemical industry, and it plays a decisive role in modernization development and construction [1,2,3,4,5,6]. When the pipeline cannot guarantee good quality in the production process and is in a harsh environment for a long time [7,8], defects occur due to the dual action of internal and external corrosion. If the corrosion is intensified, loss of life and property will be stimulated. At present, corrosion detection in the accessible area of the pipeline can be better solved by ultrasonic [9,10], magnetic flux leakage [11,12], eddy current [13,14] and other conventional detection methods. To ensure the stability and durability of the pipeline, supporting frames are set at a certain distance periodically. The corrosion detection of the support is space-sealed, and the effective detection range of common portable EMAT is limited, making it difficult to detect the point-to-point ultrasonic thickness measurement [15]. If the corroded pipeline is lifted from the support for routine detection, it is not only time-consuming and laborious, but also dangerous because of leakage. Therefore, it is necessary to provide a detection method for pipeline corrosion at the support.

Ultrasonic guide waves are typically used in pipeline corrosion detection due to their long propagation distance and small attenuation. Low-frequency ultrasonic guide waves propagating along the pipeline axis have been extensively studied due to their long distance and fast detection ability. For example, Liu et al. [16] utilized an improved planar electromagnetic array coil to successfully identify the outer diameter of 42 mm at a 320 KHz excitation T (0,1) mode. For a circumferential groove crack defect on 5 mm thick pipe and 1.5 mm deep, Wang et al. [17] set defects at different positions at the bending point of aluminum tube with outer diameter of 30 mm and 1.5 mm thickness, and conducted experiments with a periodical permanent magnet electromagnetic acoustic transducer (PPM EMAT) with an excited T (0,1) mode of 320 KHz. The results show that the defect detection effect on the outside was better than that on the inside. Liu et al. [18] used pulsed electromagnet insertion EMAT to stimulate L (0,2) mode at 220 KHz and checked the effectiveness of the sensor for quantifying pipeline defects through experiments and simulations, relative to the guided wave along the pipe axis. Another circumferential mode guide wave propagating along the circumference of the pipe applies to defect detection. For instance, Thon et al. [19] studied four PPM EMAT structures on steel pipe through experiment and simulation, and they proved that the maximum thickness loss of 50% can be observed with the SH_4_ mode of 795 KHz. Joaquin et al. [20] excited the A_0_ mode of 158 kHz on a 9.27 mm thick steel pipe and adopted intelligent voice processing technology to successfully locate and quantify the weakness with a minimum depth of 1.85 mm. Liu et al. [21] controlled the incidence angle of the sound wave by changing the polarization angle of the magnet. The simulation and experimental results showed that the oblique incidence EMAT sensor could selectively detect the defects by using an A_1_ high-order mode guided wave in the excitation plate at 2.25 MHz. Matthew et al. [22] used shear guide waves to detect the circumferential direction of the pipeline. Moreover, the accurate location of defects in pipelines has been secured, proving that this technology can quickly screen out pipelines with defects.

However, in most cases, to ensure the smooth operation of the pipeline and reduce the environmental damage to the pipeline, the edge of the support structure and the outer wall of the pipeline will be welded together, and the modal guide wave adopted in the above study produces obvious reflection when it encounters these welded supports [23]. Therefore, many researchers have explored the influence of support. For example, Pouyan Khalili et al. [24] conducted experiments on the corrosion of pipeline support parts and found that the piezoelectric probe was used to stimulate 2 MHz A_1_ modes to detect pitting defects of 1 mm depth with higher sensitivity. Shivaree K et al. [25] proved through experiments and simulation that the A_1_ mode under the 1 MHz excitation of the piezoelectric sensor could detect the defects in the hidden area of the pipeline support with a diameter of 1.5 mm and a depth of 2 mm.

The above studies show that although the piezoelectric probe can detect defects, it has high requirements on the specimen surface and the experimental repeatability is poor. The electromagnetic ultrasonic detection technology is more convenient for the quantitative detection and evaluation of defects by non-contact detection, including no coupling or surface polishing. For example, Wu et al. [26] used the finite element method to simulate the interaction between circumferential Lamb waves and lamination. Based on this interaction, defects between aluminum tube stacks can be effectively identified to improve the accuracy of quantization. Compared with Lamb waves, the shear horizontal (SH) wave vibration direction is parallel to the axial direction of the pipeline, insensitive to the surface of the specimen, and the mode of SH wave is simple at a low frequency, which is conducive to the detection of defects in the support. For instance, Nicholas et al. [27] found that the support structure has little influence on the propagation of SH waves, so the 730 kHz SH_1_ mode excited by EMAT was adopted in the experiment. The thinning area inside the support structure has been successfully detected. However, none of the above studies focused on the defects hidden behind the support structure. To detect such defects, it is necessary to cross the support structure twice, which has a significant impact on the signal and requires further analysis and research.

In this paper, a low-order circumferential shear horizontal (CSH) wave is adopted to detect the perforated defects hidden in the pipe support. Firstly, the propagation law of waves at different structures is simulated, and then the influence of different size defects and different types of defects, supports and weld structures on ultrasonic propagation are compared using experiment and simulation, and in-depth analysis and discussion are carried out.

## 2. Theory

### 2.1. SH Wave Propagation Characteristics

It is assumed that the surface area of the specimen is infinite, and the material of the specimen is isotropic. When the SH guide wave propagates in the specimen, the upper and lower surfaces of the specimen are free, and the dispersion equation of SH guide wave can be obtained through the boundary conditions:(1)$$\frac{\omega^{2}}{c_{s}^{2}}-\frac{\omega^{2}}{c_{p}^{2}}={\left(\frac{n\pi }{2h}\right)}^{2}$$
are symmetric and antisymmetric modes, and when *n* are even and odd, respectively. By defining the angular frequency *ω*, plate thickness *h* and shear wave velocity *c_s_* in the dispersion Equation (1), the phase velocity *c_p_* and group velocity *c_g_* of SH guide waves in specimens can be obtained, and the approximate dispersion curve of 3 mm thick steel plate is obtained, as shown in Figure 1.

Further analysis of the particle motion equation of SH guide wave can be obtained:(2)$$u_{y}^{s}=B\cos\left(n\pi z/d\right)\cos\left(kx-\omega t\right)$$
(3)$$u_{y}^{a}=A\sin\left(n\pi z/d\right)\cos\left(kx-\omega t\right)$$
where $u_{y}^{s}$ and $u_{y}^{a}$, respectively, are symmetrical and antisymmetric modal particle displacement, *A* and *B* are arbitrary constants that are not zero, and *k* is the wave number. Figure 2 shows the wave structure diagram of the SH_0_ mode and SH_1_ mode. It can be seen that when *n* is 0, SH_0_ mode vibrates uniformly in the direction of the entire plate thickness. When *n* is 1, particle displacement directions of the upper and lower surfaces of SH_1_ mode are opposite. Based on the above analysis, among all the modes excited by PPM EMAT, only the SH_0_ mode at low frequency has a simple structure; it has no dispersion and mode conversion, and the particle displacement is parallel to the surface of the specimen. Therefore, when testing the specimen with supporting structure, SH_0_ mode has a certain degree of advantage.

### 2.2. PPM EMAT Energy Exchange Mechanisms of Ferromagnetic Materials

The working mechanism of EMAT in ferromagnetic materials is mainly composed of Lorentz force, magnetostrictive force, and magnetization force [28]. Among them, the magnetization force generated by the drastic changes in the electromagnetic field on the material surface only exists on the material surface and is far smaller than other forces, so it is ignored [29].

The PPM EMAT energy exchange principle studied in this paper is shown in Figure 3, where the magnetic sensing line of the periodic array permanent magnet forms a static magnetic field *H* along the *y* direction, and the magnetic field of the periodic array exists alternately in the up and down directions. The current *i* is passed into the racetrack coil under the permanent magnet, resulting in a current density of *J_e_*. This will produce a dynamic magnetic field *H_d_*, due to the influence of the dynamic magnetic field. Then, close to the transducer side of the specimen surface, skin depth will be induced by eddy current *J_e_*, which is opposite to the racetrack coil on both sides of the eddy current direction. The specimen in static magnetic field *H*, dynamic magnetic field *H_d_* and eddy current *J_e_* under the common influence will form in the direction of the same Lorentz force *F_L_* in the x direction. When the particle inside the specimen feels the action of Lorentz force *F_L_*, it will generate vibration in the *x* direction, and then form SH guided waves propagating bidirectionally along the *z*-axis. The force *F_L_* will be considered equal to:(4)$$F_{L}=F_{Ls}+F_{Ld}=J_{e}\times \left(H+H_{d}\right)$$
where *F_Ls_* are the static forces and *F_Ld_* are the dynamic forces, *J_e_* is the eddy current induced by the coil, and *H* and *H_d_* are static and dynamic magnetic fields, respectively.

Considering that the tested material is steel, the magnetostrictive effect should also be considered [30]. Taking the left bias static magnetic field as an example, the bias static magnetic field *H_Ly_* acts vertically on the linear part of the racetrack coil, and the static magnetic field *H_Ly_* acts together with the tangential component of the dynamic magnetic field *H_d_*_1*x*_ and the normal component *H_d_*_1*y*_ to produce a synthetic magnetic field *H_LT_* with an angle of *β*_1_ with the *Y*-axis. The size of the angle *β*_1_ changes with the size of the dynamic magnetic field, and the specimen expands and deforms along the direction of the synthetic magnetic field *H_LT_*. Generally, the bias magnetic field intensity provided by the permanent magnet in PPM EMAT is much greater than the dynamic magnetic field intensity, that is, *H_Ly_* >> *H_d_*_1*x*_, *H_d_*_1*y*_; *β*_1_ ≈ 0. Therefore, magnetostrictive force *F_M_* can be ignored, Lorentz force is the main ultrasonic source generated by SH guided wave, and force *F* generated by EMAT can be calculated as:(5)$$F=F_{L}+F_{M}\approx F_{L}$$

## 3. Finite Element Simulation

### 3.1. Model Building

Finite element simulation was carried out based on COMSOL Multiphysics software. First of all, through the analysis of the dispersion characteristics and structure of CSH waves in the theoretical section, the pure CSH_0_ wave with an excitation lower than the modal cutoff frequency of CSH_1_ is selected, so the size of the periodic array magnet is set as 10 mm × 3 mm × 10 mm (length × width × height). In the simulation, the transducer consists of six pairs of periodic array magnets with a racetrack coil. Secondly, to reduce the calculation cost, Q235 steel pipe (*ρ* = 7850 kg/m^3^, E = 210 GPa, *ν* = 0.30) with a length of 50 mm, outer diameter of 165 mm and thickness of 3 mm is used as the waveguide, and both sides of the pipe were set as a low reflection boundary to prevent interference to detection. Meanwhile, to simulate an H-shaped welded bracket, a steel plate model of the same material 50 mm in length, 30 mm in width and 3 mm in thickness was set on the pipe surface to simulate the propagation law of one side of the bracket. In the finite element modeling of the contact surface between the bracket and pipe, under ideal conditions, the contact between two surfaces was assumed to have a smooth surface [31], but it is worth noting that the ideal weld almost has no echo. To simulate the influence of the weld on the detection in the experiment, the density and Young’s modulus of the weld material were increased in the simulation [32]. Finally, the transducer was set on the outer surface of the pipe with an excitation frequency of 500 KHz (corresponding to 1.5 MHz·mm in a 3 mm thick steel pipe, which is lower than the cutoff frequency of the CSH_1_ mode). The model of the cross-pipe support structure is shown in Figure 4, where Figure 4a is the 3D simulation model and Figure 4b is the schematic diagram of a circular structure.

Two types of defects are set behind the support structure of the pipe model, namely three crack defects with a length of 7, 5, 3 mm and a width of 2 mm, and one round hole defect with a diameter of 5 mm. The defects are all through holes, and the middle line of the excitation and receiving sensors are aligned with the defects. Through the voltage received by the receiving sensor, the detection performance of the CSH_0_ mode across the support can be evaluated. To ensure the accuracy of the calculation results, the maximum cell size of the specimen was set as 1/6 of the wavelength, and the time step was set as less than 1/6 of the period. The excitation signal was a five-period Hanning window modulated sine wave with a fixed amplitude of 20 V.

### 3.2. Crossing Pipe Support Structure Simulation

Figure 5 depicts the sound field of PPM EMAT inside a 3 mm thick steel pipe at 1.5 MHz·mm, where Figure 5a,b are guided wave propagation observations cross the support and weld structure from different angles at 35 μs and 45 μs, respectively. The CSH_0_ wave is excited in both circumferential directions of the transducer. The maximum displacement amplitude of each guided wave crossing the support structure is further extracted, and the propagation characteristics of CSH_0_ mode guided wave across the support structure are explored by a particle displacement program. When the particle displacement on the opposite side reaches 0.849 mm, the maximum displacement of particles without crossing the support is 0.845 mm at 35 μs, representing a mere 0.5% decrease compared to that of the opposite side with almost negligible loss; at 45 μs, the guided wave can be divided into three parts: the reflected echo from the support structure, the guided wave within the support structure, and the guided wave across the support structure. The particle displacement of the reflected echo from the support is 0.347 mm, while that of the guided wave propagating within it is 0.451 mm, representing 40.9% and 53.1% of their respective amplitudes on opposite sides. The particle displacement of the guided wave across the support is 0.529 mm, which is 62.3% of the amplitude on the other side, indicating a reduction in energy by 37.7% after crossing.

To investigate the impact of the support structure, a simulation was conducted on a welded pipe with identical frequency. Additionally, by taking the particle displacement on the other side of 0.849 mm as a reference, at 35 μs, it was consistent with the propagation law across the support structure, and there was almost no energy loss; at 45 μs, the guided wave is divided into two parts: the reflected echo of the weld and the guided wave across the weld, in which the particle displacement of the reflected echo of the weld is 0.133 mm, which is 15.7% of the amplitude on the other side, and the particle displacement of the guided wave after crossing the weld is 0.687 mm, which is 80.9% of the amplitude on the other side, indicating that the energy after crossing is weakened by 19.1%. The results show that, compared with the weld structure, the support structure will not only reflect the guide wave but also disperse the energy of the guided wave, which greatly increases the difficulty of defect detection.

## 4. Experimental System Construction

After the simulation of the cross-pipe support capability of the CSH_0_ mode guided wave, Figure 6 shows the PPM EMAT excitation and receiving of an experimental system of a CSH_0_ mode guided wave, which is mainly composed of a computer, an electromagnetic ultrasonic defect detector, PPM EMAT and a steel pipe. The sensor used in the experiment has the same size and structure as the transducer used in the finite element simulation in Section 3 and is composed of a periodic array magnet and a racetrack coil. The permanent magnet is made of NdFeb material, the remaining flux density is 1.4 T, the geometric size (length × width × height) is 10 mm × 3 mm × 10 mm, and the racetrack coil is a double-layer printed circuit board (PCB) coil with 2 mm inner diameter, 17 mm outer diameter, and 30 turns. In the experiment, the relative positions of the excitation sensor and the receiving sensor are consistent with those of the simulation. The excitation sensor is driven by the five-period Hamming window modulation signal and excises the CSH_0_ mode on the circumferential direction of the steel pipe (outer diameter 165 mm, thickness 3 mm, length 1 m). The computer is responsible for inputting the generated excitation signal into the electromagnetic ultrasonic defect detector. After the electromagnetic ultrasonic defect detector is modulated, the signal is received by the receiving sensor, and the data are processed by the electromagnetic ultrasonic defect detector and finally fed back to the computer.

The support structure of the experimental pipe is welded from a steel plate of the same material, which is 1 m long, 30 mm wide and 3 mm thick. After the support structure, defects of different types and sizes are designed, among which the crack lengths are 3 mm, 5 mm and 7 mm, respectively, and the width is the same. The diameter of circular hole defects is 5 mm, and all defects are through holes. The axial distance of each defect is 200 mm, and the structural diagram of its specific size and position is shown in Figure 7. At the same time, for comparison, the support structure was replaced by a welded steel pipe in another identical pipe.

## 5. Results and Discussion

### 5.1. Study on Defects of Different Sizes Detected by Crossing Pipe Support Structure

Figure 8a shows the normalized signals of three simulated crack defects. Figure 8b shows the Gaussian-filtered signals of three experimentally obtained crack defects. The right side of Figure 8 presents enlarged images of each defect signal. Revised sentence: A comparison reveals that both simulation and experiment exhibit three distinct wave packets, namely the support weld echo (wave packet I), the bottom of the support echo (wave packet II) and the defect echo (wave packet III), The support structure will not only reflect the waves, but also the waves in the support plate will be received due to reflection after contacting the bottom-end face, resulting in finally receiving three wave packets. This finding is consistent with the simulation results in Figure 5a.

The CSH_0_ wave utilized in this study is generated by EMATs consisting of a periodic array magnet and racetrack coil. The magnet design should satisfy the relationship among the phase velocity, excitation frequency and wavelength, which is:(6)$$d=\frac{\lambda }{2},\lambda =\frac{C_{p}}{f}$$
where *d* is the width of the magnet, *λ* is the wavelength, *C_p_* is the phase velocity, and *f* is the operating frequency. According to the relative position of the defect and the transducer in Figure 4b, it can be seen that the propagation distance is 346 mm and the group velocity of CSH_0_ wave is 3290 m/s. It is calculated that the time to receive the defect echo should be about 105 μs. In the experiment, the defect region is 95~110 μs, which is consistent with the calculated results.

Subsequently, the detection of crack defects of different sizes across the support in simulation and experiment was studied and peak signals were extracted and normalized. The results are shown in Table 1 and an amplitude comparison diagram was drawn as shown in Figure 9. As can be seen, the law of the simulation and experiment is the same. When the defect size increases, the detection amplitude of defects after the support increases, indicating that this method can effectively detect defects after the support. When the size of the defect decreases, the sensitivity of the detection also decreases. The experimental signal of the 3 mm crack defect is small, which is because the welding method of the supporting structure on the steel pipe is spot welding, resulting in the connection of the support not being tight, and the gap in the weld will affect the actual detection. Moreover, the 3 mm crack defect is relatively small. The defect echo of the guide wave signal will be further weakened after it passes through the support structure twice, so the defect signal cannot be effectively received during the experiment.

### 5.2. Research on Different Types of Defects Detected across the Support Structure

In the process of conveying media, the pipeline will not only produce crack defects under the joint action of tensile stress and specific corrosive media and certain pressure and temperature, but it will also exhibit pitting corrosion under the action of media, air, and local stress coupling. Pitting corrosion with the extension of time will lead to the penetration of the steel pipeline, forming hole defects.

Therefore, to study the characteristics of circular hole defects and crack defects in detection, a circular hole defect with a diameter of 5 mm was set behind the simulated support structure in the study. Compared with a 5 mm crack defect of the same size, the results are shown in Figure 10, where the left side is the propagation cloud image of CSH_0_ mode at the defect, the middle is the simulated received signal, and the right side is the amplified signal at the defect. Defect signal amplitudes of Figure 10a,b were extracted, respectively. The back amplitude of the crack defect was 0.078 V, and that of a circular hole defect was 0.042 V. The amplitude of the circular hole defect was about two times lower than that of the crack defect. By comparing the propagation cloud image of the CSH_0_ mode of the crack defect and the circular hole defect, it can be seen that when the guided wave meets the plane defect of Figure 10a, the reflection and diffraction directions of the wave are concentrated and the echo energy is concentrated, so the received reflected echo energy is strong. When it comes to the camber defect of Figure 10b, the CSH_0_ wave will reflect and diffract when interacting with the defect, resulting in a more chaotic propagation direction and more dispersed echo energy compared with the plane surface, and effective echo cannot be received.

Based on the simulation results, the propagation characteristics of cracks and circular hole defects were further studied by experiments, and the detected signals were obtained as shown in Figure 11. The left side is the experimental received signal, and the right side is the amplified signal diagram of the defect. Defect signal amplitudes of Figure 11a,b were extracted, respectively. The back amplitude of the crack defect was 0.111 V, and that of the circular hole defect was 0.039 V. Compared with simulation, the signal amplitude of the circular hole defect was three times lower than that of crack defect in the experiment, which was not much different from simulation results. However, the signal amplitude gap of detecting defects with radian is larger than that of planar defects, making it more difficult to detect. The main reason for this phenomenon is that many uncontrollable noise signals will be generated during the experiment, which will cause interference with the experimental signal, and the smaller the amplitude of the back wave, the more easily the defect signal will be submerged in the noise. This results in the post-processing process cannot obtain the ideal signal. Therefore, in the actual detection, it is necessary to distinguish different defects carefully. When the amplitude of the back wave of the defect is very small, the defects existing behind the support structure may not only be small crack defects, but may also be large diameter circular hole defects.

### 5.3. Cross Pipe Support and Weld Structure Comparative Study

To further verify the influence of the support structure on CSH_0_ wave propagation, a welded structure was set instead of the support structure for detection in the experiment. The main difference between the weld structure and the support structure was that the influence of the support steel plate was missing in the detection. The specific test results are shown in Figure 12. Figure 12a shows the crack detection signals of different sizes across the weld structure. By comparing Figure 12a and Figure 8b, it can be seen that the received signals that cross the weld structure lack the bottom of the support echo (wave packet II), showing only the weld echo (wave packet I) and the defect echo (wave packet III). The defect echo signal is in the range of 95~110 μs and its signal amplification diagram is shown in Figure 12b.

Results of the crossing support structure in Figure 8b and across the weld structure in Figure 12a were normalized. The ratio of peak defect signal to the calculated support/weld amplitude is shown in Table 2, and the comparison diagram was drawn according to Table 2, as shown in Figure 13. The influence of the two structures on the propagation of CSH_0_ mode can be analyzed through the support/weld amplitude ratio. The amplitude ratios of 3, 5, and 7 mm defects are 0.55, 0.81 and 0.93, respectively. It can be observed that regardless of the detected defect size, the support/weld amplitude ratio is always less than 1. This shows that the support has a stronger influence on signal propagation than the weld. With the decrease in the defect size, the support/weld amplitude ratio gradually decreases, indicating that the influence of the support steel plate gradually increases, resulting in the reduction of the received echo. The main reason for this gap is that when the guided wave passes through the weld of the welded steel pipe, the weld will simply reflect it, without dispersing the energy of the guided wave. On the way, the guided wave passes through the support structure, and part of the energy of the guided wave is reflected by the weld of the support, while another part of the energy is propagated inside the support structure after passing through the weld, which seriously weakens the energy of the guided wave after crossing the support. As a result, defects cannot be accurately located and quantified during detection, which corresponds to the simulation results in Section 3.2 and verifies the correctness and effectiveness of the simulation results.

## 6. Conclusions

A crossing support structure defect-detection method based on the PPM EMAT method was proposed, and the effectiveness was verified by experiments. The conclusions are drawn as follows:(1)Based on the excitation principle of PPM EMAT, a circumferential detection simulation system was established for the three-dimensional crossing-pipe support structure. By comparing and exploring the interaction between the CSH_0_ mode and the support or weld structure, it was found that the guided wave signal would be shunted into the support plate when passing through the support, but the CSH_0_ mode hardly changed. Clear echoes of cracks of different sizes are received, which proves the effectiveness of this method in pipe detection.(2)The set-up of the test experiment of the crossing-pipe support structure was as follows. Firstly, simulation and experimental rules of defects of different sizes were compared to verify the rationality of the simulation results. Secondly, the interaction between different types of defects and CSH_0_ mode were compared. Through the comparison of simulation and experimental signals, it was found that when detecting circular hole defects, the detection amplitude is reduced by about two to three times compared with crack defects, due to the reflection and diffraction effect of the camber surface. Finally, the defect-detection signals of the support and weld structures were compared, and it was found that the support structure can produce an obvious inhibition effect on the detection of small defects.(3)By analyzing the experimental and simulation results, it was found that the low-order CSH_0_ mode can effectively detect crack defects of more than 3 mm, which provides a basis for the large-area defect-detection research of the supported pipeline. However, when the defect size is smaller or a hole defect is detected, due to the influence of the supporting structure, the detection of the defect will be hindered, and the sensor designed in the study is not suitable for pipeline detection in a high-temperature environment. Therefore, subsequent research needs to focus on the optimization of such defects and the influence of temperature change on the structure and performance of the sensor.

## Figures and Tables

**Figure 1 sensors-23-05535-f001:**
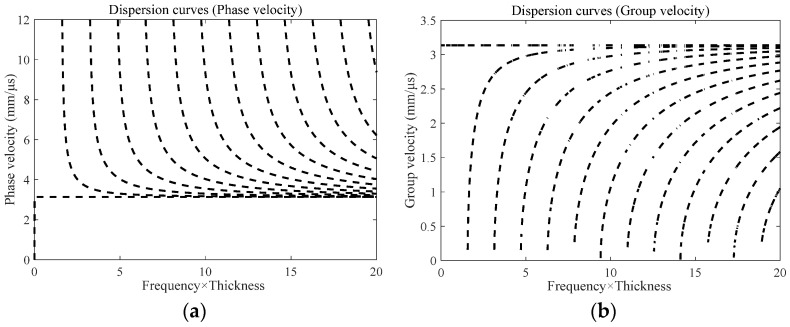
Scatter curve diagrams of SH guide plate. (**a**) Phase velocity dispersion; (**b**) Group velocity dispersion curve.

**Figure 2 sensors-23-05535-f002:**
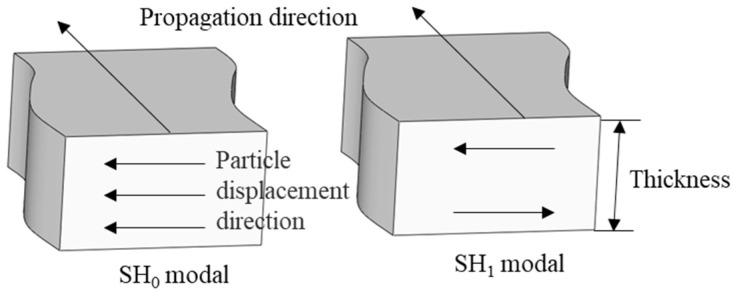
Modal wave structure diagrams of SH_0_ and SH_1_.

**Figure 3 sensors-23-05535-f003:**
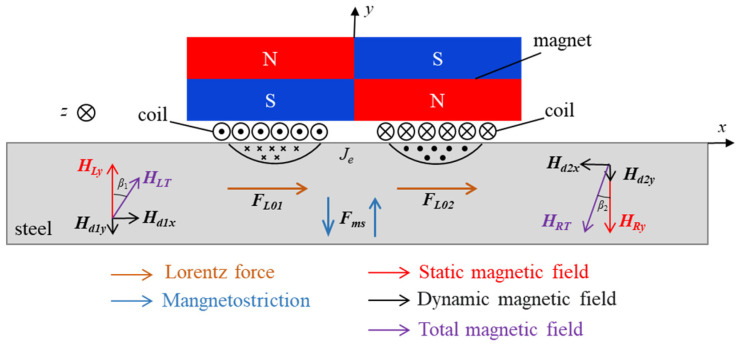
Excitation principle of PPM EMAT.

**Figure 4 sensors-23-05535-f004:**
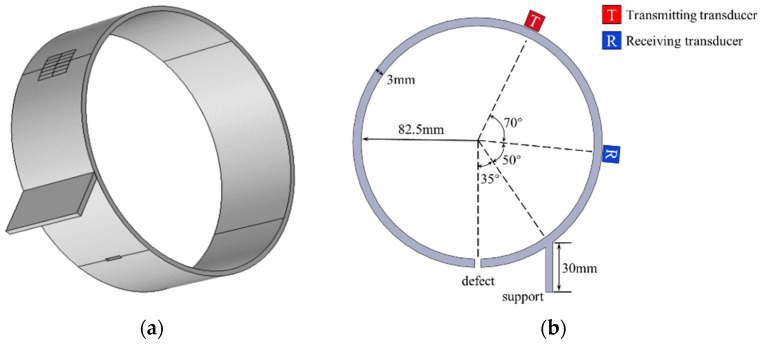
Schematic diagrams of the pipe structure. (**a**) 3D model; (**b**) Structural diagram.

**Figure 5 sensors-23-05535-f005:**
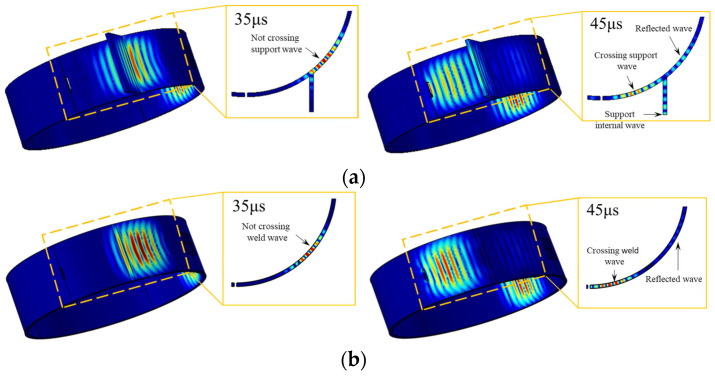
Sound fields at different structures. (**a**) support structure; (**b**) weld structure.

**Figure 6 sensors-23-05535-f006:**
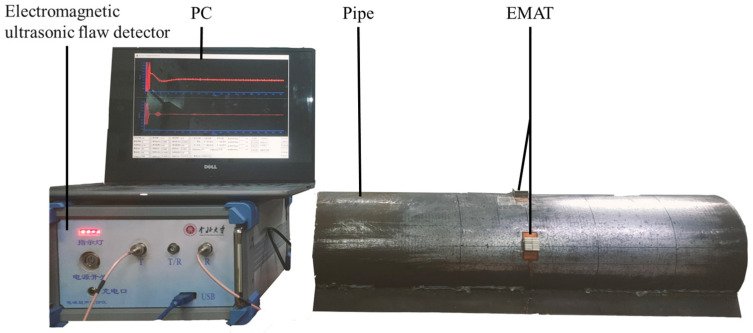
Experimental system diagram.

**Figure 7 sensors-23-05535-f007:**
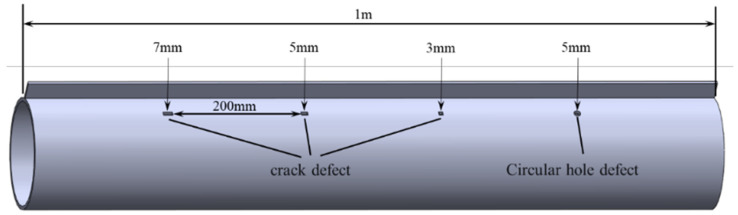
Schematic diagrams of pipe structure.

**Figure 8 sensors-23-05535-f008:**
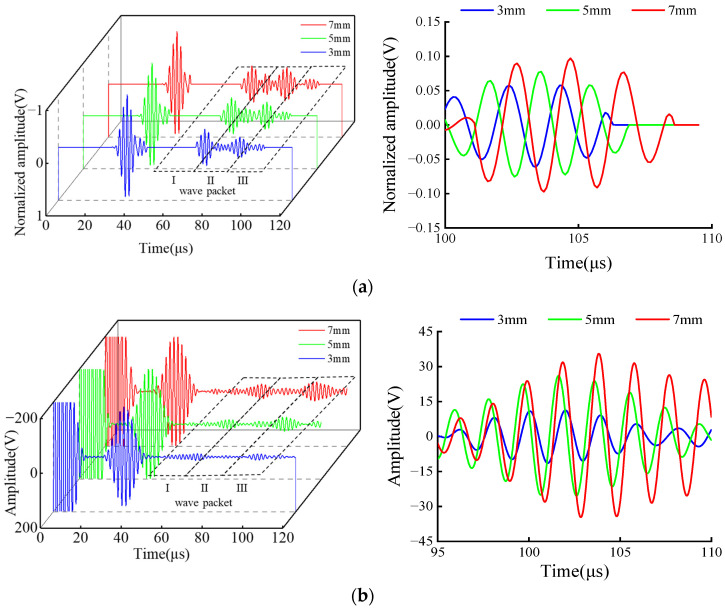
Signal diagram. (**a**) Simulation; (**b**) experimental.

**Figure 9 sensors-23-05535-f009:**
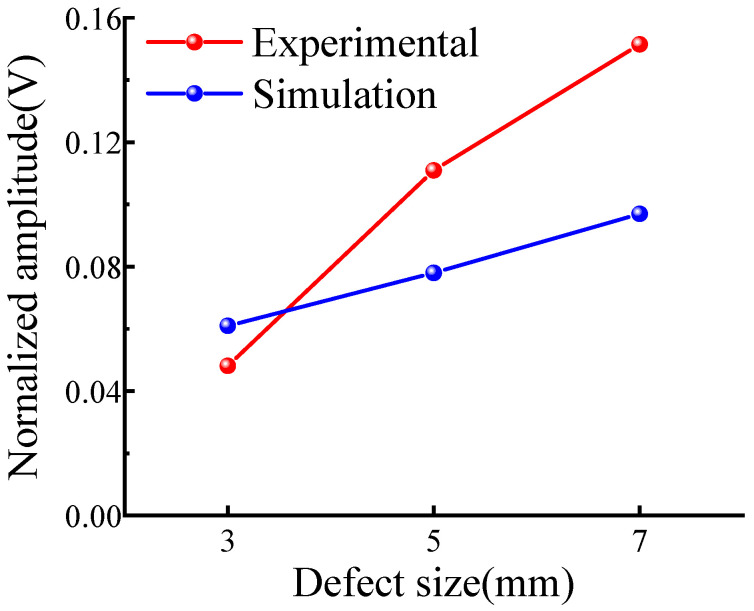
Comparison of support simulation and experiment.

**Figure 10 sensors-23-05535-f010:**
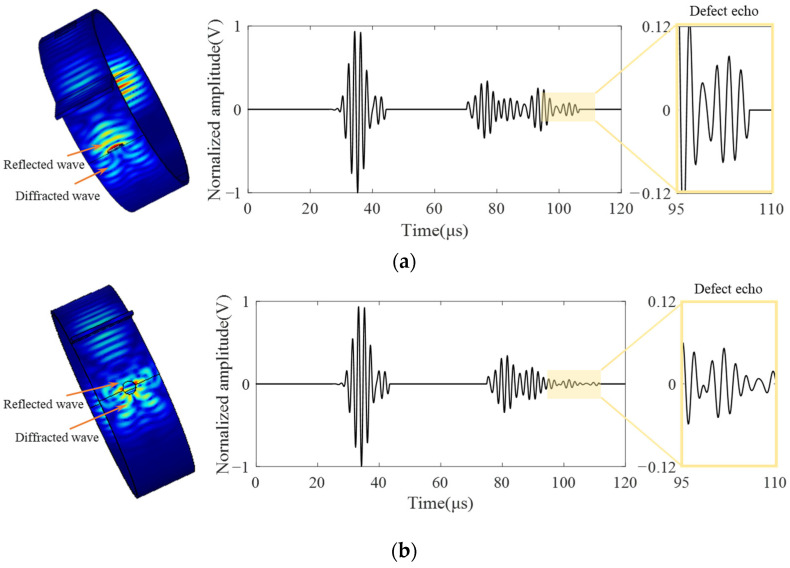
Simulation result. (**a**) crack defect; (**b**) circular hole defect.

**Figure 11 sensors-23-05535-f011:**
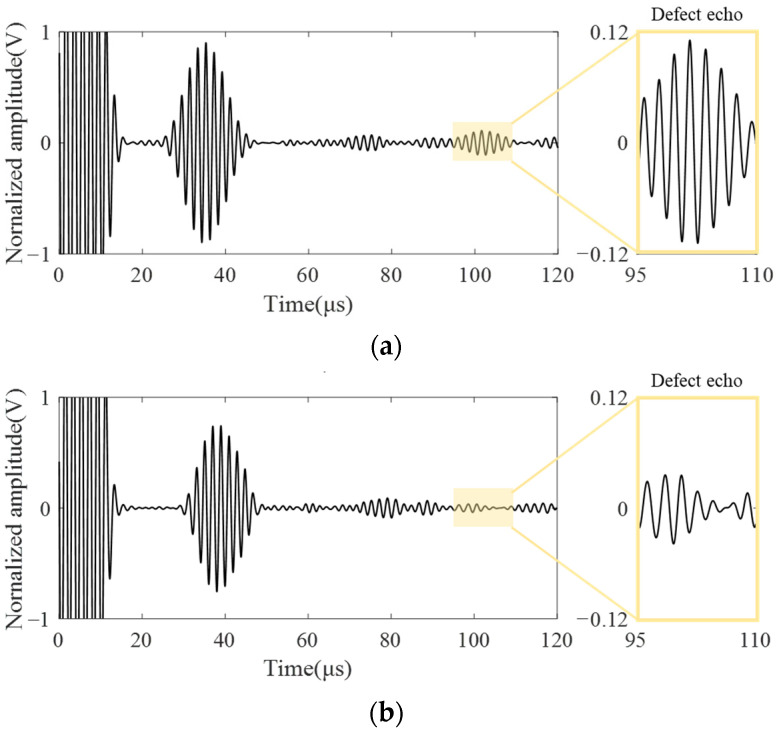
Experimental result. (**a**) crack defect; (**b**) circular hole defect.

**Figure 12 sensors-23-05535-f012:**
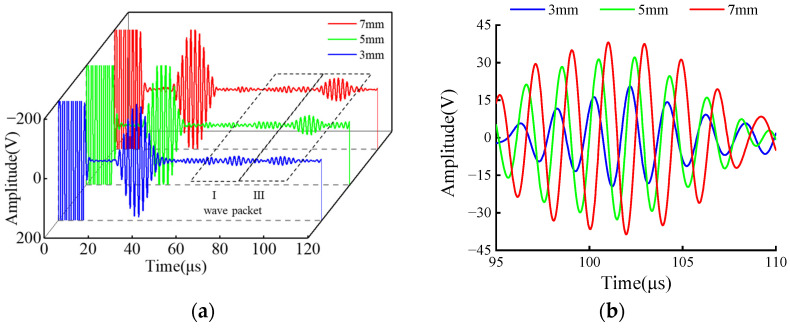
Defect detection diagrams cross the weld structure. (**a**) Crack detection signals cross weld structure; (**b**) Magnification of defect signal.

**Figure 13 sensors-23-05535-f013:**
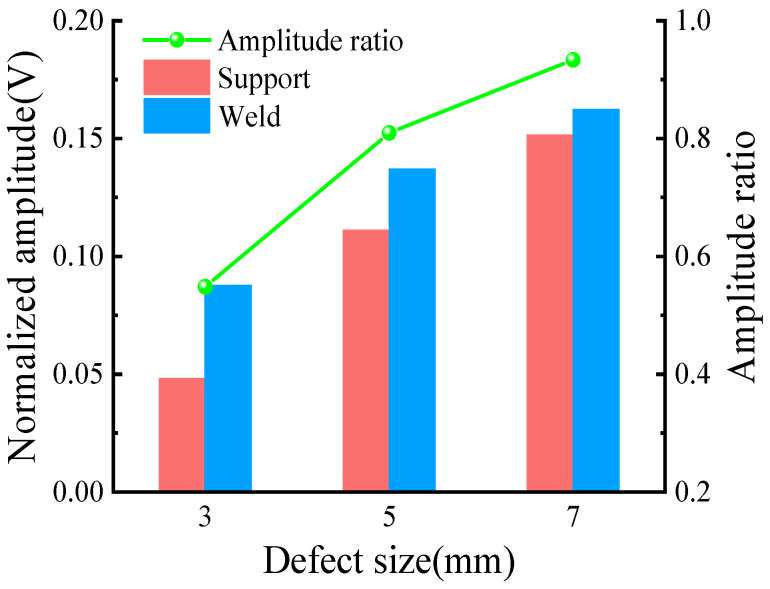
Comparison of different structures.

**Table 1 sensors-23-05535-t001:** Normalized amplitude of defect echo in simulation and experiment.

Defect Size (mm)	3	5	7
Simulation amplitude (V)	0.061	0.078	0.097
Experimental amplitude (V)	0.048	0.111	0.152

**Table 2 sensors-23-05535-t002:** Normalized amplitude of defect echoes of different structures.

Defect Size (mm)	3	5	7
Cross the support structure (V)	0.048	0.111	0.152
Cross the weld structure (V)	0.088	0.137	0.162
Support/Weld amplitude ratio	0.55	0.81	0.93

## Data Availability

Data sharing not applicable.

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
