# Peer review of "Research on Crossing-Pipe Support Structure Defect Detection of EMAT-Excited CSH Wave"

_sensors, 2023, doi:10.3390/s23125535_

Round 1

Reviewer 1 Report

The authors presented an interesting paper. The paper is well-written and can be useful for practice. The research design, questions, hypotheses, and methods are clearly stated. The paper is suitable for the Journal of Sensors.

Minor review:
1. Please ensure the abstract is short but reflects the approach, results, and conclusions correctly and concisely.

2. Please check the keywords to ensure they are appropriate and complete.

3. All variables used in the equations should be clearly explained in the text.  

4. In references, the DOI should be added for individual sources if possible.

5. The references and the citation don't work the way the authors just added to the text.

6. The limitations of the proposed study need to be discussed before the conclusion.

I suggest accepting the paper, but my comments should be resolved.

Reviewer 2 Report

In this work an across support structure defect detection based on PPM EMAT method was proposed and the effectiveness was verified in experimentally. This is well prepared work, giving some original results of importance. I recommend this work after a minor revision. Some comments are listed below.

1.     The authors used abbreviated words throughout the paper, need to give the complete details of these words.

2.     The PPM-EMAT inspection could be used to detect all defects in a weld? Give explanation

3.     PPM-EMAT is more sensitive than MS-EMAT, why? Which method is more convenient.

4.     Does PPM-EMAT method can bring reduction in the structural and high simulation cost? How? Give details.

5.     Some related works are suggested to be cited, such as: J. Mater. Chem. C 2022, 10, 7469-7475.

6.     What is effect of temperature variation on this method? Temperature is directly related to the ultrasonic velocity.

7.     Provide comparison table of the current work with respect to the previously used methods.

Reviewer 3 Report

This manuscript used low-order CSH0 mode to detect the perforated defects hidden in the pipe supporting structure, which is relatively novel. This research is important to improve the ability of crossing pipe support structure inspection. However, following suggestions should be clarified before publication.

1. What is the difference in detection results if piezoelectric ultrasonic transduceris used?

2. Do the upper and lower cracks affect the current crack detection when CSH0 wave is used? How should the distance between cracks be chosen to avoid this effect?

3. Does the width of the pipe set in the simulation model affect the signal?

4. The summary section needs to be further refined, as in " The results show that this method has a good detection effect on the defects hidden behind the support, and the echo of the detection of circular hole defects is reduced by about 2-3 times compared with the crack defects, indicating that the defects with radian are more difficult to detect."

5. Part of the references is incomplete and needs to be improved.

The English needs to be fully polished.
